# Osmotic Stress Leads to Significant Changes in Rice Root Metabolic Profiles between Tolerant and Sensitive Genotypes

**DOI:** 10.3390/plants9111503

**Published:** 2020-11-06

**Authors:** Maya Matsunami, Kyoko Toyofuku, Natsumi Kimura, Atsushi Ogawa

**Affiliations:** 1Faculty of Agriculture, Iwate University, 3-18-8 Ueda, Morioka 020-8550, Japan; mayanami@iwate-u.ac.jp; 2Department of Biological Production, Akita Prefectural University, Akita 010-0195, Japan; kyotoyo@akita-pu.ac.jp (K.T.); atsu10429@gmail.com (N.K.); 3Japan Science and Technology Agency, Core Research for Evolutionary Science and Technology Project, Tokyo 102-0076, Japan

**Keywords:** metabolomics, osmotic stress, *Oryza sativa* L., rice, root

## Abstract

To breed osmotic stress-tolerant rice, the mechanisms involved in maintaining root growth under osmotic stress is important to elucidate. In this study, two rice (*Oryza sativa* L.) cultivars, IR 58 (stress-tolerant cultivar) and Basilanon (stress-sensitive cultivar), were used. After 1, 3, and 7 days of −0.42 MPa osmotic stress treatment induced by polyethylene glycol (PEG) 6000, root metabolomes were analyzed, yielding 276 detected compounds. Among 276 metabolites, 102 metabolites increased with the duration of the stress treatment in IR 58 roots, and only nine metabolites decreased. In contrast, 51 metabolites increased, and 45 metabolites decreased in Basilanon roots. Principal component analysis (PCA) scores clearly indicated differences between the cultivars and the treatments. Pathway analysis showed that the metabolites exhibiting stress-induced increases in IR 58 were those involved in sugar metabolism (such as sucrose 6’-phosphate, glucose 1-phosphate), polyamine and phenylpropanoid metabolisms (such as spermine, spermidine, gamma-aminobutyric acid (GABA)), and glutathione metabolism (such as glutathione, cysteine, cadaverine). IR 58 roots showed an increase in the most proteinogenic amino acids such as proline, serine, glutamine and asparagine. It was also maintained or increased the tricarboxylic acid (TCA) cycle intermediates (citric acid, cis-Aconitic acid, isocitric acid, fumaric acid, malic acid) under osmotic stress compared with that under control. Therefore, IR 58 actively synthesized various metabolites, and the increase in these metabolites contributed to the maintenance of important biological functions such as energy production and antioxidant defense to promote root development under osmotic stress.

## 1. Introduction

Rice is the world’s most important food crop and a primary food source for more than half of the world’s population [1]. More than 90% of the world’s rice is grown and consumed in Asia, where 60% of the world’s people live. Rice accounts for 35%–60% of the calories consumed by 3 billion Asians. Osmotic stress (drought stress and salt stress) is the most important abiotic stress for rice production. Water deficits commonly occur in rain-fed rice production areas, which account for about half of the total rice production area globally [2]. Rice productivity in rain-fed ecosystems is lower than that under irrigated cultivation because of the high sensitivity of rice to drought [3]. Therefore, further genetic improvement, in addition to novel production technologies, is required for sustainable rice production.

Abiotic stresses affect plants at morphological and agronomical (plant size; height, tiller number, biomass, root and leaf developments, yield components; grain number, grain size, sterility), physiological (reduced photosynthesis, transpiration, stomatal conductance, water use efficiency, nutrient uptake), biochemical (accumulation of osmolyte, sugars, polyamines, antioxidants, synthesis of abscisic acid), and molecular levels, thereby affecting yield. Recent progress in omics analysis has allowed us to clarify the response of various plants under stress through comprehensive quantitative analysis of genes and substances. The major abiotic stress factors, such as drought, salinity, extreme high/low temperature, cause significant changes in the composition of the plant metabolome [4,5]. Metabolomic analysis has been used in studies of rice shoot [6,7,8], and understanding the roles of low-molecular-weight metabolites in biological processes is essential for crop improvement. Suzuki et al. [6] comprehensively investigated metabolite changes in transgenic rice plants with increased or decreased Rubisco content and revealed that C and N metabolisms were widely affected. Yamakawa and Hakata [7] comprehensively investigated changes in metabolites in rice caryopsis under high temperature stress and revealed that a high temperature increased the accumulation of sucrose and pyruvate/oxaloacetate-derived amino acids and decreased levels of sugar phosphates and organic acids involved in glycolysis/gluconeogenesis and the tricarboxylic acid (TCA) cycle, respectively.

It has been said that “roots are the key to a second green revolution” [9], and deeper understanding and improvement of roots may contribute significantly to future food production [10]. By using the Rice Diversity Research Set of germplasm (RDRS), which was developed by the National Institute of Agrobiological Science (NIAS) [11], it becomes possible to compare varietal differences of rice with wide genetic variation. There are reports on root morphology and responsiveness under various stresses. Uga et al. [12] analyzed anatomical and morphological traits under rainfed upland conditions and found differences in root characteristics between japonica and indica accessions. Matsunami et al. [13] reported genotypic differences in biomass production in shoots and roots under soil moisture deficit using the RDRS. In our previous studies, the genotypic variations to affect the osmotic stress tolerances and morphological characteristics of the root system were shown among 59 cultivars of rice seedlings, including 54 cultivars in RDRS [14]. Root growth was maintained in cultivars whose shoot growth was maintained under osmotic stress. The root-system morphology of stress-tolerant cultivars was investigated in detail. L-type lateral root growth, which were longer and thicker, was maintained in stress-tolerant varieties. These results suggested that the maintenance of root-system development, especially L-type lateral roots under osmotic stress, involves genetic variation in the genes responsible for the dry matter production. The morphological response to osmotic stress in the roots has been elucidated, but the comprehensive changes in metabolites have been largely unclear.

In this study, we conducted a metabolome analysis of the roots of rice in order to compare the responses of stress-tolerant and -sensitive cultivars, which were selected from RDRS in our previous study [14], grown under osmotic stress conditions. The objective of this study was to identify metabolites and their underlying pathways associated with osmotic stress tolerance, that is, metabolites that show a distinct response to osmotic stress and to clarify their genotypic differences.

## 2. Results

### 2.1. Overview of Osmotic Stress on the Root Metabolome and Cultivar Differences

The two cultivars showed clear differences in their shoot and root biomass production responses to polyethylene glycol (PEG)-induced stress conditions during 7 days of stress treatment (Figure 1). Basilanon showed a larger reduction in shoot dry weight (DW) under stress than IR 58; the percentage of reduction in shoot DW relative to the control treatment was 18% in IR 58 and 56% in Basilanon. The root DW of IR 58 was maintained under osmotic stress, whereas that of Basilanon was significantly decreased (by 58%). Thus, seven days of stress treatment resulted in physiological changes related to differences in biomass production between the two cultivars. Sucrose content as a respiratory substrate greatly increased in the IR 58 roots under stress (Table 1). Osmotic stress significantly increased fructose content in IR 58 but significantly decreased it in Basilanon (Table 1).

Therefore, we focused on the metabolomic changes occurring within roots during stress. From the roots collected after 1, 3, and 7 days of osmotic stress treatment, 276 compounds (148 cation and 128 anion compounds) were detected. The number of compounds changed by osmotic stress differed by time series and cultivar (Table 2). By 1 d, osmotic stress increased 52 and 71 compounds in IR 58 and Basilanon, respectively. Over time, the number of metabolites that increased under osmotic stress increased in IR 58 (83 and 102 compounds at 3 and 7 d after stress), whereas the number of increased compounds was slightly decreased in Basilanon (64 and 51 compounds). In IR 58, there were fewer compounds (1,3, 9 compounds at 1, 3, and 7 d of stress, respectively) that decreased under stress treatment compared to Basilanon. By contrast, in Basilanon, the number of compounds that were decreased by stress treatment rose as the duration of stress treatment increased; there were 5, 15, and 45 compounds at 1, 3, and 7 d of stress, respectively. These results indicate that more substances were increased by stress in IR 58 and, conversely, more substances were decreased by stress in Basilanon.

PCA scores also indicated differences between cultivars and treatments (Figure 2A). The first principal component (PC1) explained 25.7% of the total variability, and clearly separated the stress treatment on the positive side from the control on the negative side. The PC1 score became positive as the stress period increased. PC2 explained 17.3% of the variability and separated the cultivars, placing Basilanon on the positive side and IR 58 on the negative side. In IR 58, the PC2 scores were comparable between treatments, whereas the score was higher in the stress treatment for Basilanon.

The PC1 loading ranged from −0.08 to 0.12, and the top absolute values were those of β-Ala (0.12), hypotaurine (0.11), 3-aminoisobutyric acid (0.11), O-acetylserine (0.11), and cGMP (0.11) (Appendix A). In addition to proteinogenic amino acids, many amino acids and their intermediates and/or derivatives, such as β-Ala, O-acetylserine, and N6-Methyllysine, showed high PC1 loadings. The PC2 loading ranged from −0.12 to 0.13; the absolute values were high for mucic acid (0.13), glucaric acid (0.12), glucoaminic acid (0.12), UDP-glucuronic acid (−0.12), and pyridoxine (0.12). For most amino acids, the PC1 loadings were high and positive (Figure 2B). The PC2 loadings for amino acids were not large compared to those of PC1. However, some metabolites in the pathway of glycolysis and the citric acid cycle showed high absolute values for PC2 loading, which included both positive and negative values depending on the metabolite (Figure 2C).

### 2.2. Amino Acid Changes under Osmotic Stress

Since the observed increases in the levels of amino acids under stress were remarkable, we investigated the abundances of proteinogenic amino acids to determine whether they changed during osmotic stress. The concentrations of glycine (Gly), serine (Ser), proline (Pro), and cysteine (Cys) were significantly increased by stress treatments regardless of cultivar (Figure 3A). In addition to these amino acids, valine (Val), threonine (Thr), asparagine (Asn), aspartic acid (Asp), lysine (Lys), glutamic acid (Glu), glutamine (Gln), methionine (Met), histidine (His), phenylalanine (Phe), and tyrosine (Tyr) were significantly increased by the stress treatment in the IR 58 roots. Changes in amino acid concentrations during stress varied by amino acid and cultivar. (Figure 3B). In general, the levels of amino acids in the stressed condition were higher than those in the control. Amino acids, such as Gly and Asn, were higher in the early stages and decreased gradually, while others, such as Ser, Thr and Cys, increased and maintained a higher level than the control during the entire stress period. The amino acid concentrations in roots under stress were generally higher in IR58 than in Basilanon, with some exceptions, such as Ser and Gly.

### 2.3. Genotypic Differences in Osmotic Stress-Induced Metabolome Pathways in Roots

The effects of osmotic stress on the major metabolome pathways are shown in Table 3. In the starch and sucrose pathway, significant increases in sucrose 6-phosphate, UDP-glucose, and glucose 1-phosphate under stress conditions were observed in IR 58, whereas only sucrose 6-phosphate increased, and the other metabolites tended to be decreased by stress in Basilanon. In the glutamine (Gln) and glutamic acid (Glu) pathways, Gln, Glu, and gamma-aminobutyric acid (GABA) were increased by stress in IR 58. No compounds from this pathway increased in Basilanon. The stress treatment increased the levels of many metabolites in the glutathione metabolism pathway; glycine (Gly), cysteine (Cys), glutathione, γ-Glu-Cys, and NADP were increased significantly regardless of the cultivar. The increase in Cys was remarkable; the stress/control ratio was 15 in IR 58 and 17 in Basilanon. In addition to these compounds, spermine, cadaverine, Glu, and spermidine were increased in IR 58. In the polyamine metabolic pathway as well as in the glutathione metabolism pathway, metabolites of IR 58 were significantly increased by stress treatment. Only N-acetylputrescine increased in Basilanon, while spermidine and s-adenosylmethionine significantly decreased. In the phenylpropanoid synthesis pathway, which leads to lignin synthesis, phenylalanine (Phe), tryptophan (Try), and spermidine were increased by stress in IR 58. In fact, the lignin content in the IR 58 roots was significantly increased by stress treatment (Table 4). In the glycolysis pathway, glucose-1-phosphate (G1P), 3-phosphoglyceric acid (3PG), and 2-phosphoglyceric acid (2PG) were significantly increased by stress treatment in IR 58, whereas phosphoenolpyruvic acid (PEP) was increased in Basilanon (Table 3). Almost all the metabolites in the tricarboxylic acid (TCA) cycle in Basilanon were remarkably decreased by stress, except for acetyl-CoA.

### 2.4. Changes in Metabolites in the TCA Cycle

As we found cultivar differences in the abundance of TCA cycle metabolites (Figure 2C, Table 3), we further investigated the changes in the metabolites in the TCA cycle and its upstream metabolites in the roots (Figure 4). As shown in Table 3, the abundance of phosphoenolpyruvic acid (PEP) under stress was larger than that of the control in Basilanon roots, and this trend started at 3 days after treatment (DAT). There was no clear difference in the amount of pyruvic acid among cultivars or treatments. Acetyl CoA increased with stress at 3DAT in IR 58 and then decreased to a level comparable to that of the control at 7DAT. The amount of acetyl CoA in Basilanon at 7DAT was higher than that in the control. Interestingly, IR 58 and Basilanon exhibited clearly different metabolic responses to stress in terms of the metabolites generated after entering the TCA cycle under stress. For most of the metabolites in the TCA cycle, IR 58 control and stress treatments and the Basilanon control treatment exhibited similar changes in the amount by which the levels of these metabolites increased between 3DAT and 7DAT, except for 2-oxoglutaric acid (2OG), whereas these metabolites remained low in the roots under stress. Thus, the cultivar difference in response to stress in the TCA cycle was distinct from that of glycolysis. Citric acid synthase (CS) activity, which catalyzes the conversion of acetyl-CoA and oxaloacetate into citrate and H-SCoA, was determined (Figure 5). The CS activities per unit of fresh roots were comparable between both cultivars in the control. The activity significantly increased with stress in IR 58, whereas there was no significant change in stressed Basilanon.

## 3. Discussion

### 3.1. Maintenance of Root Development and Increased Sucrose under Stress in Tolerant Cultivars

The principal aim of the present study was to identify metabolites that are associated with osmotic stress tolerance in roots. Therefore, we assessed differences in the response to osmotic stress at the metabolite level in stress-tolerant and -susceptible rice cultivar roots. The biomass production of shoots and roots showed that the tolerant cultivar IR 58 maintained higher dry weight, especially root weight, under osmotic stress than Basilanon (Figure 1). Shoot growth is dependent upon the nutrient supply from the roots. On the other hand, roots rely on photosynthetic products provided by shoots. Both shoot and root performances are important for the maintenance of material production under stress. In this experiment, it was found that, under osmotic stress, IR 58 minimized losses of its aboveground biomass by maintaining root development and thereby maintained its ability to gain photosynthetic products for the development of roots (Figure 1). Our previous study characterized the morphology of IR 58 roots under stress and found that IR 58 maintained the root system under osmotic stress by enhancing the number of L-type lateral roots, which are longer and thicker lateral roots [14]. Sucrose has been reported to accumulate in plants and plays important regulatory functions in stressed plants [5,15]. It has been reported that sucrose is involved in the development of lateral roots that occupy most of the root system [16,17]. The increase in sucrose abundance at the roots of IR 58 under stress suggests that sucrose serves as an energy source or plays a role in osmoregulation for the maintenance of root development in stress-tolerant cultivars [18,19].

### 3.2. Amino Acids Were Major Metabolites Increased in Roots under Stress

Both proteinogenic amino acids and non-proteinogenic amino acids have been reported to accumulate in plants under abiotic stress conditions [4,5,20]. Β-Alanine, which showed the highest PC1 loading (Appendix A), is a non-proteinogenic amino acid. Β-Alanine has important roles in plant physiology and metabolism, directly acts as a defense compound under various abiotic stresses, and indirectly affects other processes because it is a precursor for the compounds pantothenate and CoA, which are involved in a variety of functions [21]. Furthermore, the polyamines spermine/spermidine, propionate, and uracil have been shown to be precursors of β-alanine in plants. Β-alanine is converted to the osmoprotective compound β-alanine betaine in some species and converted to the antioxidant homoglutathione in others [21].

With regard to proteinogenic amino acids, significant increases in many amino acids (Gly, Ser, Pro, Val, Thr, Cys, Asn, Asp, Lys, Glu, Gln, Met, His, Phe, and Tyr) under osmotic stress were observed in the roots of IR 58 (Figure 3). Pro has been a target for metabolic engineering of stress tolerance in many plant species [15]. High salinity, drought, and cold stresses regulate several protein kinases and alter Pro levels to enhance stress tolerance [22], and Pro also plays an important role as a compatible osmolyte [23,24]. Accumulation of Pro under stress conditions plays an important role in serving as a sink for excess reductants, providing the NAD+ and NADP+ necessary for the maintenance of respiration and photosynthetic processes [25]. Studies using knock-out or over-expression plants of Pro synthesis-related genes in *Arabidopsis* and/or rice showed a positive relationship between Pro levels and stress tolerance under salt or osmotic stress induced by PEG [26,27,28]. In addition to Pro, the concentrations of many amino acids increased under stress conditions, including Ser and Thr, which have been reported to increase in response to stress and which have demonstrated roles in chemical defense against abiotic stresses and in plant growth and development, cell division, and phytohormone regulation [29,30]. Although the absolute amount was small, Cys, like Pro, did not accumulate under control conditions, but accumulated markedly under stress (Figure 3), suggesting it has specific roles under stress conditions. The accumulations of Asn, Asp, Glu, and Gln were markedly enhanced in roots of IR58 plants under osmotic stress. Asn and Gln are closely related to carbon and nitrogen metabolisms and are involved in the synthesis of materials necessary for plant growth and in the transport and storage of nitrogen. Because of its high stability and N/C ratio, asparagine is required not only for protein synthesis but also for long-term storage and long-distance transport of nitrogen. Plants rapidly convert inorganic nitrogen to amino acids to avoid the toxicity of ammonium ions, and glutamine synthetase and glutamate synthetase function in this process [31]. In tomato, exogenous spermidine treatment relieved nitrogen metabolic disturbances caused by salinity-alkalinity stress and eventually promoted plant growth [32]. The increased polyamines in IR 58 roots under stress may support nitrogen metabolism, resulting in high accumulation of these amino acids (Table 3).

### 3.3. Elevated Substances in Various Metabolic Pathways are Associated with Varietal Differences in Stress Tolerance

In IR 58, the number of metabolites that increased under stress rose as the duration of stress treatment increased (Table 1). By contrast, in Basilanon, fewer metabolites increased, while the number of metabolites that decreased rose as the period of stress treatment increased. The PCA of all 276 compounds clearly indicated a difference between cultivars and treatments (Figure 2B,C, Appendix A). Metabolome analysis revealed clear differences between cultivars in terms of their metabolite accumulations in response to osmotic stress and suggested that promoting metabolism more than under non-stress conditions was necessary to maintain root development under stress.

The pathway analysis shown in Table 3 clearly showed the cultivar difference in pathways involved in antioxidative stress (starch and sugar metabolic pathways, glycolytic pathways, amino acid synthesis metabolic pathways, phenylpropanoid metabolic pathways, and glutathione metabolic pathway metabolites) [33] as well as in the polyamine metabolic pathways, which are related to cell growth and proliferation [34,35] (Table 3). GABA is a non-proteinogenic amino acid accumulating under various stresses, and have suggested many important roles such as osmoregulation, signaling and others [36,37]. In an experiment comparing the amount of GABA in stress-tolerant and -sensitive sorghum leaf, it was reported that the increase in GABA under stress was observed only in the tolerant cultivar and was similar between the control and stressed cultivar in the sensitive cultivar [38]. Although the target of the sorghum study was leaf GABA content, our root study also shows the significant increase in GABA only in the tolerant cultivar, suggesting that the increase in GABA in the plant body was effective as a characteristic of the stress-tolerant genotypes.

Polyamines are also well-studied metabolites involved in plant responses to various abiotic stresses [39]. Polyamines have a wide range of functions and have shown that they are closely associated with plant growth, nucleic acid and membrane structural stabilities, and stress tolerance. It has been pointed out that the role of polyamines in stress conditions varies among plant species, and exogenous polyamine (putrescine) improves seed germination and the growth of alfalfa under osmotic stress conditions [40]. It has also been reported that an *Arabidopsis* mutant, which cannot produce spermine, was hypersensitive to salt and drought, suggesting the importance of spermine under stress [41]. In this study, polyamines such as spermine and spermidine accumulated in IR 58 roots (Table 3), suggesting the polyamines may have contributed to osmotic tolerance in rice roots. Another polyamine, putrescine, was not significantly increased under stress. Spermine and spermidine are synthesized from putrescine [42]; therefore, it was suggested that the demand for each polyamine influenced the accumulation of other polyamines.

Lignin accumulation occurs as a result of osmotic stress conditions [43,44], such as that occurring under salt, drought, or metal stress conditions [23,45]. Stress treatment increased the lignin content of roots only in IR 58 (Table 4). In IR 58, the phenylpropanoid metabolic pathway was significantly elevated under osmotic stress (Table 3). Phenylpropanoid synthesis is induced by stress [46]. The phenylpropanoid metabolic pathway leads to the lignin synthesis pathway [47]. Furthermore, in IR 58, the proline content was significantly increased under osmotic stress (Figure 3). In tobacco leaves, increased proline metabolism due to osmotic stress increases the production of secondary metabolites and activates the shikimate pathway, resulting in increased lignin biosynthesis [48]. This difference was considered to be the mechanism that affects the difference in lignin content between the two varieties and the difference in stress tolerance.

### 3.4. Maintenance of the TCA Cycle May Be Key to Stress Tolerance

Respiration synthesizes ATP during the oxidation of carbon compounds and generates energy necessary for biological activities. In this study, we found an interesting cultivar difference through the pathway from sugar to glycolysis and then to the TCA cycle. In the glycolytic pathway, there was no clear difference in response to stress in either cultivar; however, stress decreased the levels of metabolites from the initial steps in the TCA cycle in Basilanon, whereas those of IR 58 were comparable with those of the control (Figure 4, Table 3). Therefore, we focused on the activity of CS, which catalyzes citrate from acetyl CoA and oxaloacetate, because its activity may have differed between cultivars. Our results show that CS activity in IR 58 was significantly higher under stress than under control conditions (Figure 5). CS activity is the limiting factor of the first step of the TCA cycle; thus, our results suggest that the root system of IR 58 maintained the TCA cycle by up-regulating CS activity, thereby generating energy for growth under stress. In addition to the above pathway, pyruvate dehydrogenase complex (PDC), PEP carboxylase, pyruvate carboxylase, and malate dehydrogenase are involved in the metabolism of substances entering the TCA cycle. We did not measure the activity of these enzymes in this study, but the activities of relevant enzymes may explain the high accumulation of PEP in Basilanon roots under stress. It has been reported that sucrose accumulated and organic acids in TCA cycle metabolic intermediates decreased under high-temperature stress in mature kernels of rice [7]. Metabolites representative of glycolysis and the TCA cycle, such as malate, glyceric acid, and glyceric acid-3-phosphate, were reduced at the tillering stage at the end of the dry-down treatment [8]. They also found that the root metabolites representative of glycolysis and the TCA cycle were largely reduced in the aus-type rice variety “N22”, which showed the least root elongation and highest reduction in root fresh weight under drought stress. Therefore, we suggest that the enzyme activity relevant to glycolysis and the TCA cycle may be reflected in a cultivar difference in root development under stress conditions.

## 4. Materials and Methods

### 4.1. Plant Growth and Stress Treatment

Two indica rice (*Oryza sativa* L.) cultivars ‘IR 58’ and ‘Basilanon’ were selected from our previous study as osmotic stress-tolerant (IR 58) and stress-sensitive (Basilanon) cultivars [14]. Plants were grown in a growth chamber (MLR-350H; Sanyo Ltd., Moriguchi, Japan) at 28 ± 0.2 °C with a relative humidity of 70%, and a 12 h photoperiod. The photon flux density of photosynthetically active radiation at the top of each plant was 320 μmol m^−2^ s^−1^. Before transplanting, seeds were sown in petri dishes that had been sterilized with hypochlorous acid, and then incubated at 28 °C in the dark for 3 days. Then, 20 germinated seeds were transplanted onto plastic nets that floated on a hydroponic solution in a 1000 mL glass beaker. The solution contained 1.5 × 10^−3^ M KNO_3_, 1.0 × 10^−3^ M Ca (NO_3_)_2_, 2.5 × 10^−4^ M NH_4_H_2_PO_4_, 5.0 × 10^−4^ M MgSO_4_, 1.3 × 10^−5^ M Fe-EDTA, 2.3 × 10^−6^ M MnCl_2_, 1.2 × 10^−5^ M H_3_BO_3_, 1.9 × 10^−7^ M ZnSO_4_, 7.9 × 10^−8^ M CuSO_4_, and 7.5 × 10^−9^ M (NH_4_)_6_Mo_7_O_24_. The beakers were covered with heavy paper to block the root systems from light exposure. The solution was aerated using an aerator (HP α 10000; Marukan Co., Ltd., Osaka, Japan).

Seven days after transplanting, 200 g polyethylene glycol (PEG) 6000 per liter of nutrient solution was dissolved to induce osmotic stress. The water potential of the nutrient solutions was determined using a vapor pressure osmometer (Model 5520; Wescor Instruments, Logan, UT, USA). The resulting water potential values were −0.42 Mpa (stress treatment) and −0.08 Mpa (control). The osmotic stress treatment was applied for 7 days according to our previous study [14]. At 7 days after treatment, four plants per beaker were collected and separated into shoots and roots. The samples were dried at 80 °C for more than 3 days, and then the dry weight was measured.

### 4.2. Metabolome Analysis

Root samples were collected from control and PEG treatment at 1, 3, and 7 days (24, 72, 168 h) after treatment for the measurement of metabolites. Roots were carefully washed, and excess water was removed using a paper towel in order to avoid contamination of the solution. The roots were immediately frozen in liquid nitrogen and stored at −80 °C until analysis. Metabolites were determined using Human Metabolome Technologies Inc. (Tsuruoka, Yamagata, Japan). A total of 500 microliters of methanol containing 50 μM internal standard was added to approximately 50 mg of fresh frozen samples, and they were homogenized using a cell disruptor (BMS-M10N21; Bio Medical Science Inc. BMS Tokyo, Japan) under cooling (4 °C, 1500× *g*, 2 min × 2 times). Thereafter, 500 μL of chloroform and 200 μL of Milli-Q water were added, mixed, and centrifuged (2300× *g*, 4 °C, 5 min). The aqueous layer was transferred to an ultrafiltration tube (Millipore, Ultra Free MC PLHCC HMT centrifugal filter unit, 5 kDa). It was centrifuged (9100× *g*, 4 °C, 120 min) and ultra-filtered. The filtrate was dried and dissolved in 50 μL of Milli-Q water. The samples obtained were then subjected to capillary electrophoresis time-of-flight mass spectrometry (CE-TOF-MS) analysis using the Agilent CE-TOF-MS system (Agilent, Palo Alto, CA, USA) as described by Soga et al. [49].

### 4.3. Glucose, Fructose, and Sucrose Contents

Capillary electrophoresis used in CE-TOF-MS is a method of separating by the difference in moving speed caused by the difference in physical properties such as the charge and size of the substance, so uncharged saccharides are excluded from the measurement target. Therefore, the glucose, fructose, and sucrose contents of the samples were assayed using the coupled enzymatic assay method by Guglielminetti et al. [50]. Approximately 100 mg of root sample was frozen in liquid nitrogen, ground to a powder, and then extracted with 600 µL of 5.5% HClO. The samples were stored at −20 °C for 10 min, 4 °C for 50 min, and then centrifuged (5 min, 16,000× *g*, 4 °C). The supernatant was saved, and the pellet was washed with 200 µL of distilled water and centrifuged again. The resulting supernatant was combined with the initial supernatant and 120 µL of 3.5 M K_2_CO_3_. The samples were stored at −20 °C for 10 min and then centrifuged (5 min, 16,000× *g*, 4 °C). The supernatants were incubated at 37 °C for 30 min and mixed with 1 mL reaction mixture. The mixture was again incubated at 37 °C for 30 min, and the increase in A_340_ was recorded. The composition of the reaction mixture for the glucose assay was Tris HCl (120 mM, pH 7.6), MgCl_2_ (3 mM), ATP (2 mM), NADP (0.6 mM), 1 U hexokinase (FUJIFILM Wako Pure Chemical Co., Osaka, Japan), and 1 U glucose 6-phosphate dehydrogenase (FUJIFILM Wako Pure Chemical Co., Osaka, Japan). Fructose was assayed using the reaction mixture of the glucose assay plus 2 U phosphoglucose isomerase (Sigma-Aldrich Japan, Tokyo, Japan). Sucrose was hydrolyzed using 85 U invertase in 15 mM sodium acetate (pH 4.6) (Sigma-Aldrich Japan, Tokyo, Japan).

### 4.4. Citrate Synthase Activity

Citrate synthase (CS) activity of the roots was measured according to Hayes and Ma [51] with slight modifications; 100–200 mg of frozen root sample stored at −80 °C was ground using a mortar and pestle with 1 mL of extraction buffer containing 0.1 M Tris HCl (pH 8.0), 5 mM MgCl_2_, 5 mM Na-EDTA, 10% glycerol, and 0.1% Triton X-100. Then, 5 µL of 0.5 mM phenylmethylsulphonyl fluoride (PMSF) in methanol was added and mixed well. The samples were centrifuged (15 min, 15,000 rpm, 4 °C). The supernatants were desalted using an Econo-Pac 10DG column (Bio-Rad, USA). The 50 µL of prepared solution was incubated in 1 mL of reaction buffer containing 50 mM Tris-HCl (pH 8.0), 5 mM MgCl_2_, 100 µM 5,5-dithio-bis-2-nitrobenzoic acid (DTNB), 0.2 mM acetyl CoA, and 0.5 mM oxaloacetate. The CS reaction was initialized by the addition of oxaloacetate and measured by the reduction of acetyl CoA in the presence of DTNB at A_412_.

### 4.5. Lignin Content

Root lignin content was determined according to Suzuki et al. [52] with slight modifications. Dried root samples were weighed and then pulverized with a Multi-Beads Shocker (model MB455AU (S); Yasui Kikai, Osaka, Japan). The powdered samples were extracted with 1 mL of water and centrifuged at 16,000× *g* for 10 min at room temperature (20 °C). The supernatants were discarded, and the pellets were extracted with 1 mL methanol at 60 °C for 20 min and centrifuged at 16,000× *g* for 10 min at room temperature. The supernatants were discarded, methanol extraction was repeated, and the pellets were vacuum dried. Then, 1 mL 3 N HCl and 0.1 mL thioglycolic acid (Nacalai Tesque, Kyoto, Japan) were added, and the samples were heated at 80 °C for 3 h. After centrifugation at 16,000× *g* for 10 min at room temperature, the supernatants were removed, and the pellets were vortexed for 30 s in 1 mL distilled water. After centrifugation at 16,000× *g* for 10 min at room temperature, the supernatants were discarded and the pellets were resuspended in 1 mL 1N NaOH and then shaken at 80× *g* overnight. The samples were centrifuged at 16,000× *g* for 10 min at room temperature, and then 1 mL of the supernatant was transferred to 1.5-mL tubes and acidified with 0.2 mL concentrated HCl. The tubes were centrifuged at 16,000× *g* for 10 min at room temperature after they had been chilled at 4 °C for 4 h. The supernatants were removed, and the pellets were dissolved in 1 mL 1N NaOH. After a 50-fold dilution with 1N NaOH, the solutions were measured using a UV-Vis spectrophotometer (UV-mini 1240; Shimadzu, Kyoto, Japan). The calibration curves were created using a lignin alkali solution (Sigma-Aldrich Japan, Tokyo, Japan), which was prepared as outlined above. The absorbance at 280 nm was recorded in a 10 mm quartz cell using a UV-Vis spectrophotometer. The lignin concentration was determined by dividing the amount of lignin calculated based on the calibration curve by the dry weight of the sample after methanol extraction.

### 4.6. Data Analysis

Principal component analysis (PCA) was performed using SampleStat ver.3.14 (Human Metabolome Technologies, Yamagata, Japan). The metabolite networks were constructed using the KEGG (Kyoto Encyclopedia of Genes and Genomes) pathway maps web tool (http://www.genome.jp/kegg/). *T*-tests and ANOVA were performed using JMP 8 (SAS Institute Inc., Cary, NC, USA).

## 5. Conclusions

By comparing the metabolomes of osmotic stress-tolerant and -susceptible rice cultivars growing under stress, we were able to examine the effects of osmotic stress on a wide variety of compounds, not just those substances that are already known to be important in stress response. This comparison revealed that there were large differences between cultivars in many metabolites in response to osmotic stress, allowing us to gain new insight into the osmotic stress response in rice. In this study, we found that, in the roots of the stress-tolerant IR 58 cultivar, the number of amino acids (most proteinogenic amino acids, β-Alanine, hypotaurine, etc.) increased and metabolites in TCA cycle intermediates (citric acid, cis-Aconitic acid, isocitric acid, succinic acid, fumaric acid, and malic acid) were maintained under osmotic stress. Furthermore, the synthesis pathways for metabolites involved in the metabolism of nitrogen and carbon metabolism, glutathione metabolism, and polyamine metabolism were increased in IR roots. Therefore, IR 58 actively synthesized various metabolites, and the increase in these metabolites contributed to the maintenance of important biological functions such as energy production and antioxidant defense under stress conditions. These high biochemical activities may have contributed superior maintenance of root development and hence the shoot biomass production under osmotic stress. Further studies are required to understand how metabolic enzyme function is regulated by transcriptional and post-transcriptional modifications that are necessary for the enhancement of root function under stress. Determination of underlying mechanisms for osmotic tolerance and identifying key molecules as indicators for breeding criteria is essential to develop stress-tolerant varieties.

## Figures and Tables

**Figure 1 plants-09-01503-f001:**
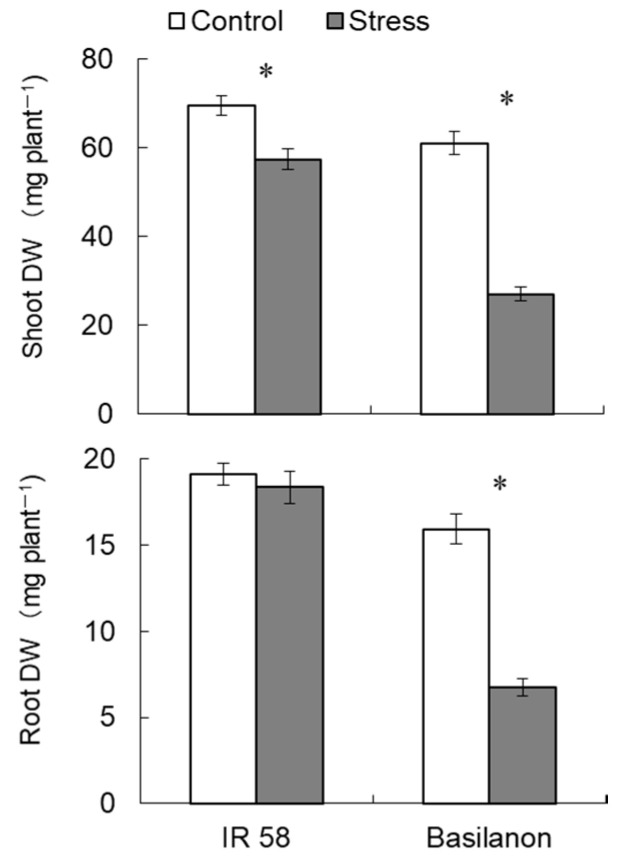
Shoot and root dry matter production of two rice cultivars in response to osmotic stress. Shoots and roots were sampled after 7 days (168 h) of treatment. Bars indicate the standard error (*n* = 12). Asterisks represent significant differences between treatments (*p* < 0.05, *t*-test).

**Figure 2 plants-09-01503-f002:**
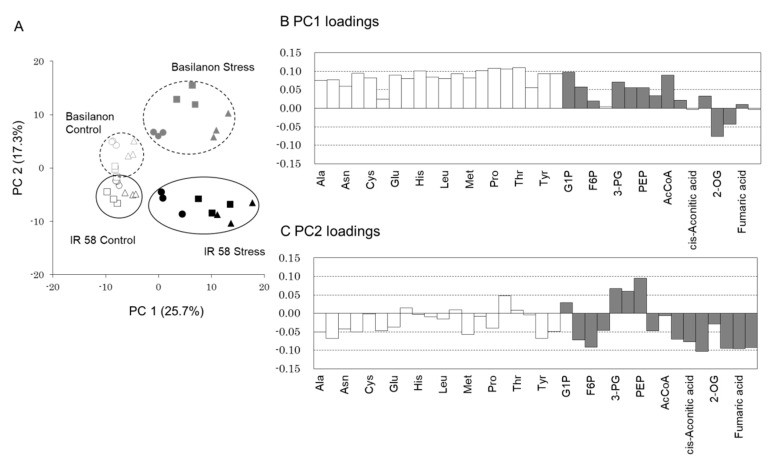
Principal component analysis (PCA) on 276 metabolites. (**A**) PCA scores are presented based on a combination of two components (PC1 and PC2). Circles, triangles, and squares indicate 1, 3, and 7 days (24, 72 and 168 h) after treatment, respectively. (**B**,**C**) Loadings of amino acid (white) and metabolites in glycolysis and citric acid cycle pathways (gray).

**Figure 3 plants-09-01503-f003:**
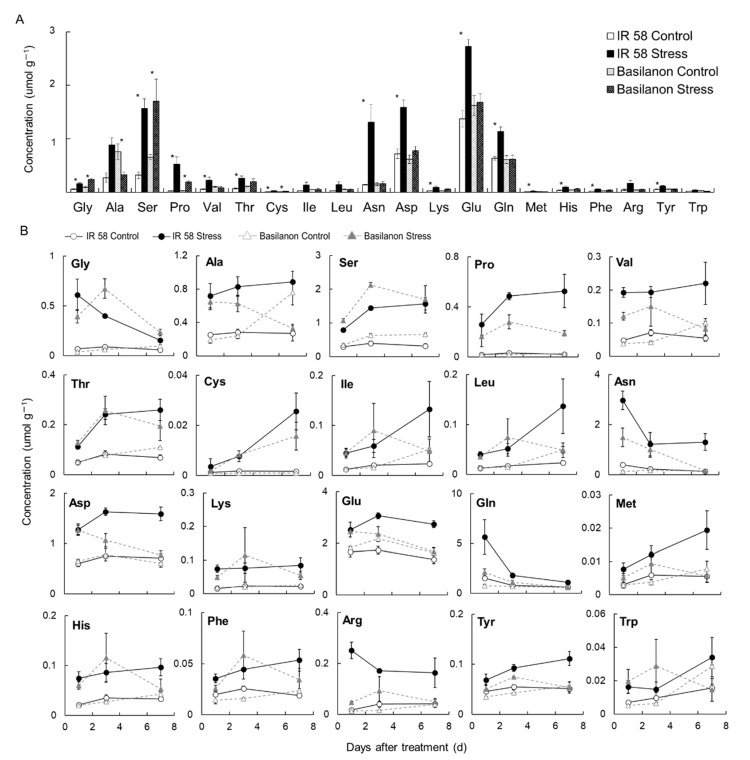
Effect of osmotic stress on root amino acid concentration in two rice cultivars. (**A**) Twenty amino acid concentration in roots at 7 days after treatment. (**B**) The changes in amino acid concentration in roots. Bars indicate the standard error (*n* = 3). An asterisk represents a significant difference between treatments in each cultivar (*p* < 0.05, *t*-test).

**Figure 4 plants-09-01503-f004:**
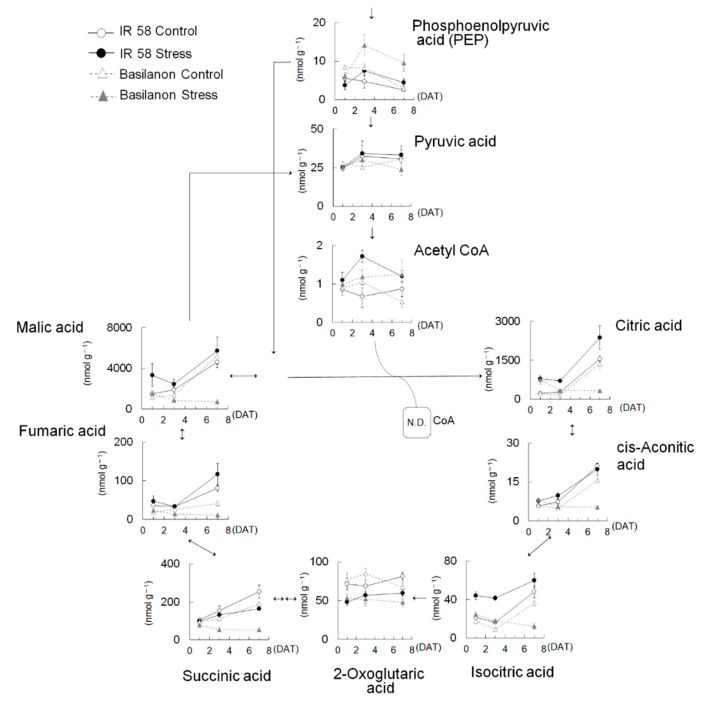
Changes in the number of metabolites in the pathways of glycolysis and citric acid cycle in the roots. Roots were sampled at 1, 3, and 7 days (24, 72, 168 h) after treatment (DAT). Bars indicate the standard error (*n* = 3).

**Figure 5 plants-09-01503-f005:**
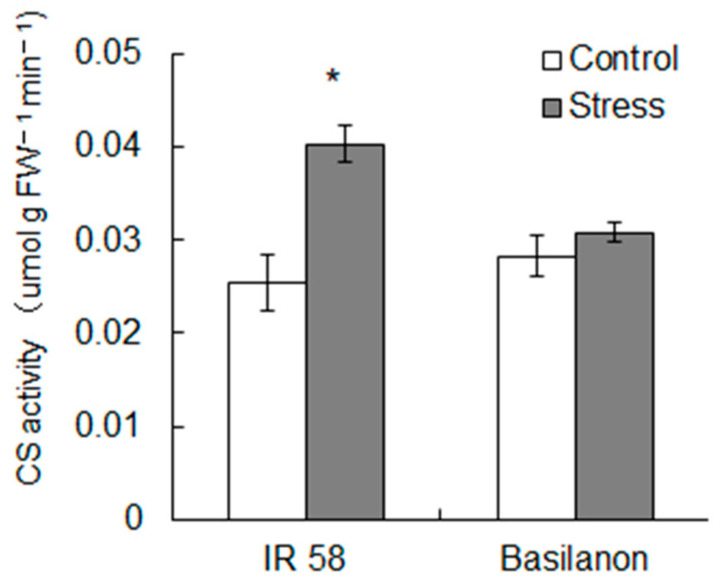
Effect of Osmotic stress on the citrate synthase (CS) activity in roots. Roots were sampled 7 days (168 h) after treatment for the determination of CS activity. Bars indicate the standard error (*n* = 5). An asterisk represents a significant difference between treatments (*t*-test, *p* < 0.05).

**Table 1 plants-09-01503-t001:** Effect of osmotic stress on sugar content in roots.

Cultivar	Treatment	Sucrose (mg g FW^−1^)	Fructose (mg g FW^−1^)	Glucose (mg g FW^−1^)
IR 58	Control	8.4b	0.4b	0.5a
	Stress	23.8a	2.3a	0.8a
Basilanon	Control	9.5ab	1.9a	0.3a
	Stress	9.2ab	0.5b	0.9a
ANOVA	Cultivar (C)	n.s.	n.s.	n.s.
	Treatment (T)	n.s.	n.s.	n.s.
	C x T	*	***	n.s.

Roots at 7 days (168 h) after treatment were sampled for the determination of sugar contents. Values followed by different letters in a column are significantly different at *p* < 0.05, according to Tukey’s test. ANOVA was performed to determine the individual and interaction effects of the cultivars and osmotic stress treatment, and *p*-values from ANOVA are presented as * and ***, indicating significance at 5 and 0.1% levels, respectively.

**Table 2 plants-09-01503-t002:** Varietal differences in the time course of increases and decreases in metabolites in roots under osmotic stress.

DAT	Cultivar	Increase	No Change	Decrease
1 day	IR 58	52	223	1
	Basilanon	71	200	5
3 days	IR 58	83	190	3
	Basilanon	64	197	15
7 days	IR 58	102	165	9
	Basilanon	51	180	45

The number of metabolites significantly increased or decreased compared to controls at 1, 3, and 7 days (24, 72 and 168 h) after treatment (DAT) by *t*-test (*p* < 0.05) among 276 compounds detected by capillary electrophoresis time-of-flight mass spectrometry (CE-TOF-MS) analysis.

**Table 3 plants-09-01503-t003:** Effect of osmotic stress on the root metabolites in a stress-tolerant rice cultivar (IR 58) and a stress-sensitive rice cultivar (Basilanon).

Pathway Cluster	Compound Name	IR 58		Basilanon	
		S/C Ratio	*p*-Value	S/C Ratio	*p*-Value
Starch and sucrose metabolism	Trehalose 6-phosphate	1.5	0.109	0.8	0.724
	Sucrose 6′-phosphate	3.1	0.002	1.5	0.005
	UDP-glucose	1.2	0.036	0.7	0.020
	UDP-glucuronic acid	1.0	0.602	0.5	<0.001
	Glucuronic acid	0.6	0.017	0.5	0.008
	Glucose 6-phosphate	1.2	0.132	0.7	0.059
	Glucose 1-phosphate	2.3	0.001	1.4	0.109
	Fructose 6-phosphate	1.0	0.813	0.7	0.046
Gln and Glu metabolism	Gln	1.8	0.005	1.0	0.924
	Glu	2.0	<0.001	1.0	0.705
	GABA	1.7	0.008	1.0	0.928
	Pyruvic acid	1.1	0.623	0.8	0.068
	Succinic acid	0.6	0.045	0.3	7.5 × 10^−4^
	2-Oxoglutaric acid	0.7	0.005	0.7	0.020
Glutathione metabolism	Gly	2.7	0.007	2.4	0.006
	5-Oxoproline	1.3	0.325	1.0	0.957
	Cys	15	0.030	17	0.042
	Ornithine	1.1	0.621	1.2	0.457
	Glutathione (GSH)	4.9	0.014	2.8	0.048
	γ-Glu-Cys	2.6	0.025	2.2	<0.001
	Spermine	3.8	0.006	0.7	0.347
	Cadaverine	4.0	0.035	1.5	0.102
	NADPH_divalent	1.6	0.171	2.0	N.A.
	Glu	2.0	<0.001	1.0	0.705
	NADP^+^	1.5	0.011	2.9	0.011
	Acetyl CoA_divalent	1.4	0.090	2.3	0.066
	Spermidine	1.8	0.013	0.4	0.007
	Putrescine	1.2	0.282	1.3	0.299
Polyamin metabolism	Agmatine	1.5	0.331	0.8	0.501
	Spermine	3.8	0.006	0.7	0.347
	GABA	1.7	0.008	1.0	0.928
	Spermidine	1.8	0.013	0.4	0.007
	Putrescine	1.2	0.282	1.3	0.299
	*S*-Adenosylmethionine	1.4	5 × 10^−4^	0.9	0.044
	*N*-Acetylputrescine	2.6	0.016	5.3	0.009
	Norspermidine	0.4	0.026	2.1	0.514
Phenylpropanoid metabolism	Phe	2.8	0.026	1.5	0.238
	Tyr	2.2	0.009	0.9	0.680
	Spermidine	1.8	0.013	0.4	0.007
Glycolysis	Fructose 1,6-diphosphate	0.7	0.073	0.8	0.428
	Pyruvic acid	1.1	0.623	0.8	0.068
	Glucose 6-phosphate	1.2	0.132	0.7	0.059
	Glucose 1-phosphate	2.3	0.001	1.4	0.109
	Dihydroxyacetone phosphate	1.1	0.483	0.7	0.060
	Fructose 6-phosphate	1.0	0.813	0.7	0.046
	Lactic acid	1.1	0.581	2.2	0.077
	3-Phosphoglyceric acid	1.5	0.012	1.5	0.116
	2-Phosphoglyceric acid	1.4	0.008	1.5	0.237
	Phosphoenolpyruvic acid	1.8	0.061	3.3	0.032
TCA cycle	Citric acid	1.5	0.080	0.2	0.010
	Malic acid	1.2	0.284	0.13	0.010
	Isocitric acid	1.2	0.100	0.3	<0.001
	*cis*-Aconitic acid	0.9	0.449	0.3	<0.001
	Fumaric acid	1.5	0.143	0.3	0.008
	Acetyl CoA_divalent	1.4	0.090	2.3	0.066
	Succinic acid	0.6	0.045	0.3	<0.001
	2-Oxoglutaric acid	0.7	0.005	0.7	0.020

Roots at 7 days (168 h) after treatment were sampled for metabolite determination. S/C ratio means the ratio of stress/control of each metabolite abundance. White letters with a black background indicate a metabolite that was significantly increased by osmotic stress, and black letters with a gray background indicate a compound that was significantly decreased by osmotic stress (*t*-test, *p* < 0.05).

**Table 4 plants-09-01503-t004:** Effect of osmotic stress on root lignin contents.

Cultivar	Treatment	Lignin Contents (mg g DW^−1^)
IR 58	Control	131.8 b
	Stress	176.0 a
Basilanon	Control	90.9 b
	Stress	121.0 b

Roots at 7 days (168 h) after treatment were sampled to determine lignin contents. Values followed by different letters in a column are significantly different at *p* < 0.05, according to Tukey’s test (*n* = 5).

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
