# Peer review of "Osmotic Stress Leads to Significant Changes in Rice Root Metabolic Profiles between Tolerant and Sensitive Genotypes"

_plants, 2020, doi:10.3390/plants9111503_

Round 1
Reviewer 1 Report
The evaluated manuscript describes a significant problem regarding drought stress and differences in the response of rice plants mainly at the level of primary metabolites. The article is well planned and written, although the editing of many figures and tables included in the text raises reservations. (1) Figure 2 and 3 size and lettering are too small; (2) Tab. S1 and Tab. 3, why is it in the text of the manuscript, moreover, the lettering is too small, illegible.
It is unclear where the quantitative data for glucose, fructose and sucrose in the results come from. Were they counted on the basis of the CE / MS results, or from additionally performed analyzes with other methods for sugars (see Materials and Methods). This should be clarified. In Tab. 4 are presented results of lignin contents, what it means there [%]?
Reviewer 2 Report
Dear Editor and authors
Thank you for the opportunity to review the article presented by Matsunami et al., "Osmotic stress leads to significant changes in rice root metabolic profiles", which represents an important scientific contribution to understanding the metabolomic changes of a species of great global importance such as rice, in the coping with a stress condition which is increasingly common such as osmotic stress (drought and salinity). The study and its methodological approach is equally of high quality, while representing a methodological challenge given the use of capillary electrophoresis as a methodology rather than other more commonly used tools (HPLC-MS-TOF), which at least I found novel. Therefore, I recommend the acceptance of the document after a series of moderate corrections that I listed below.
Title. It is extremely broad if it is not limited to the experimental design in which osmotic stress contrasting tolerance genotypes were used (I assume that cultivars were developed by selection under drought conditions). So, I suggest a minor change as “Osmotic stress leads to significant changes in rice root metabolic profiles among genotypes of contrasting tolerance” or similar.
L35, change by (drought and salt stresses)
L41, the morphological levels here mentioned can be included as yield components to give a most “agronomical” value to the study.
L43, I am not sure if the ABA content it is actually physiological or biochemical trait…
L59, please change study by studies, research or reports (there are two cited studies)
L64. Please, take into consideration that here you used PEG as the artificial medium to induce the “osmotic stress”. Therefore, the plants were growing under “osmotic stress” that was caused by PEG. I consider that it is incorrect to mention “PEG stress”, since the actual effect is the osmotic stress. Please, change “PEG stress” by “Osmotic stress” throughout the body text including figure captions and tables.
L69, please, take off cultivar (it is redundant).
L71-75. The writing is redundant, please re-do the sentences.
L83, in the figure legend, please change samples by sampled.
L87, Table 1. Please give more information about the table. You must consider that figures and tables must be self-explanatory.
L91, …interaction effects of the cultivars and osmotic stress treatment…
Table 2. Please unify the header and foot in the table since there is similar information twice. I assume that the increase and decrease reported as a result of software is including the “significant” changes.
Figure 2. What time is reported? Al the times as a global mean? Please, clarify.
L146, during osmotic stress.
L154, during all the stress period.
L163. Please, note that you are showing results for both rice cultivars, not only for IR 58
Table 3. Please, specify at the header of at the foot what is the meaning of S/C (stress/control ratio). Again, figures and tables must be self-explanatory.
L198. Please, take off “clearly differed between cultivars”. In the results sections you must show the results. The interpretation must be developed in the next chapter.
L248… energy source or “posses” a role in osmoregulation…
L266-267. I guess it is “most” adequate the word “osmolyte” than “solute”.
L271. Please, specify the type of stress.
L277… were markedly enhanced in roots of IR58 plants under osmotic stress…
L309. I guess the correct concept here is “osmotic tolerance” or “tolerance to osmotic stress”, nor “resistance”.
L359. Is it correct the middle dash?
4.4. Methods for CSA. I guess it is important a brief detail about the volumes used in the procedures.
L414 and other. Why italics?
L439… rice cultivars growing under stress…
L440… the effect of osmotic stress…
L452. Please add a final sentence about the importance of the study as support for the rice breeding in a framework of osmotic stress (drought or salinity).
References.
There are only 10 studies very updated (5 years), other 12 relatively updated (10 years) and 27 outdated (more than 10 years). A simple-fast search in Scopus using “osmotic stress” resulted in 26120 items; using “rice" as filter the results showed 6795 items; a filter with “metabolic OR metabolomics” resulted in 2501 items; other filter considering only studies from 2015 to present showed 1440. And a last filter using “Oryza” resulted in 631 items. Therefore, the references must be strongly updated to give more quality and novelty to the study. I suggest about a third of references in each category (5 years/10 years/more than 10 years).
Reviewer 3 Report
Comment 1: Please change the title. My suggestion is instead of ..."leads to significant changes...."include any other scientific terms.
Comment 2: Line no. 18 and 22, write an abbreviation for KEGG and TCA respectively.
Comment 3: Line no. 21, "osmotically-stressed" change the term
Comment 4: In the abstract, conclude with some important metabolites and their significance on rice roots
Comment 5: In key words, use a single word instead of four words as one keyword. Please check the spelling of "Or...sativa".
Comment 6: Not satisfied with the introduction section. Please improve.
Comment 7: Table 2, include some headings for the first two columns.
Comment 8: Please remove Table S1 from the main text, include as supplementary material.
Comment 9: Table 3, please follow author instructions for Table format
Comment 10: Please improve the figure 4.
Comment 11: Please avoid two or three lines as a single paragraph. Please compare results with previous research.
Comment 12: Please improve the conclusion. Please write significant amino acids/pathways and their future perspectives instead of general sentences.
Round 2
Reviewer 1 Report
The corrections proposed the first round of evaluation were introduced to the second version of the manuscript. In this situation I propose acceptance of this manuscript.
Reviewer 2 Report
Thank you for taking into consideration all the comments and suggestions.
Reviewer 3 Report
Comment 1: In title "... leads to significant changes in rice root metabolic profiles between....." looks like results and discussion. Please improve if possible.
Comment 2: Please check bond size and style and also format in Tables, particularly change Table 3.